# *In vivo* and *ex vivo* assessment of bladder hyper-permeability and using molecular targeted magnetic resonance imaging to detect claudin-2 in a mouse model for interstitial cystitis

**Nataliya Smith[1], Debra Saunders[1], Megan Lerner[2], Michelle Zalles[1,3], Nadezda Mamedova[1], Daniel Cheong[1], Ehsan Mohammadi[3,4], Tian Yuan[3,4], Yi Luo[5], Robert E. Hurst[6], Beverley Greenwood-Van Meerveld[3,4], Rheal A. Towner**[1,3,7] *

1 Advanced Magnetic Resonance Center, Oklahoma Medical Research Foundation, Oklahoma City, OK, United States of America, 2 Surgery Research Laboratory, University of Oklahoma Health Sciences Center, Oklahoma City, OK, United States of America, 3 Oklahoma Center for Neuroscience, University of Oklahoma Health Sciences Center, Oklahoma City, OK, United States of America, 4 Department of Physiology, University of Oklahoma Health Sciences Center, Oklahoma City, OK, United States of America, 5 Department of Urology, University of Iowa, Iowa City, IA, United States of America, 6 Department of Urology, University of Oklahoma Health Sciences Center, Oklahoma City, OK, United States of America, 7 Department of Pathology, University of Oklahoma Health Sciences Center, Oklahoma City, OK, United States of America

* Rheal-Towner@omrf.org

**Data Availability Statement:** All relevant data are within the manuscript and its Supporting Information files.

## Abstract

### Objectives

To determine if the URO-MCP-1 mouse model for bladder IC/BPS is associated with in vivo bladder hyper-permeability, as measured by contrast-enhanced MRI (CE-MRI), and assess whether molecular-targeted MRI (mt-MRI) can visualize in vivo claudin-2 expression as a result of bladder hyper-permeability. Interstitial cystitis/bladder pain syndrome (IC/BPS) is a chronic, painful condition of the bladder that affects primarily women. It is known that permeability plays a substantial role in IC/BPS. Claudins are tight junction membrane proteins that are expressed in epithelia and endothelia and form paracellular barriers and pores that determine tight junction permeability. Claudin-2 is a molecular marker that is associated with increased hyperpermeability in the urothelium.

### Materials and methods

CE-MRI was used to measure bladder hyper-permeability in the URO-MCP-1 mice. A claudin-2-specific mt-MRI probe was used to assess *in vivo* levels of claudin-2. The mt-MRI probe consists of an antibody against claudin-2 conjugated to albumin that had Gd-DTPA (gadolinium diethylenetriamine pentaacetate) and biotin attached. Verification of the presence of the mt-MRI probe was done by targeting the biotin moiety for the probe with streptavidin-horse radish peroxidase (SA-HRP). Trans-epithelial electrical resistance (TEER) was also used to assess bladder permeability.

**Funding:** Funding was provided by the Presbyterian Health Foundation (RAT), and the Department of Veterans Affairs (Award 1IK6BX003610-01, BGVM).

**Competing interests:** Funding was provided by the Presbyterian Health Foundation (RAT). This does not alter our adherence to PLOS ONE policies on data or materials sharing.

## Results

The URO-MCP-1 mouse model for IC/BPS was found to have a significant increase in bladder permeability, following liposaccharide (LPS) exposure, compared to saline-treated controls. mt-MRI- and histologically-detectable levels of the claudin-2 probe were found to increase with LPS -induced bladder urothelial hyper-permeability in the URO-MCP-1 IC mouse model. Levels of protein expression for claudin-2 were confirmed with immunohistochemistry and immunofluorescence imaging. Claudin-2 was also found to highly co-localize with zonula occlidens-1 (ZO-1), a tight junction protein.

## Conclusion

The combination of CE-MRI and TEER approaches were able to demonstrate hyper-permeability, a known feature associated with some IC/BPS patients, in the LPS-exposed URO-MCP-1 mouse model. This MRI approach could be clinically translated to establish which IC/BPS patients have bladder hyper-permeability and help determine therapeutic options. In addition, the *in vivo* molecular-targeted imaging approach can provide invaluable information to enhance our understanding associated with bladder urothelium hyper-permeability in IC/BPS patients, and perhaps be used to assist in developing further therapeutic strategies.

## Introduction

Interstitial cystitis(IC)/bladder pain syndrome (BPS) is a chronic inflammatory bladder health issue. This condition is predominant in females (1 in 4) and is known to lead to pain, discomfort, and tenderness in the bladder and pelvic region [1]. Although reports of IC/PBS can be traced back to the early 19[th] century [2], its symptoms are complex and multifactorial in nature [3]. Many experts believe that about 3.3 million women in the U.S. (over age 18) may have IC/BPS, as well as 1.6 million men [4, 5]. To date, there are no gold standards in the diagnosis and detection of IC/PBS and clinicians have to rule out several symptoms common to these co-morbid diseases (i.e. differential diagnosis) to begin treatment for IC/BPS [3, 6].

The cause of IC/BPS is unknown, but abnormalities in the leakiness or structure of the lining of the bladder may play a role in the development of IC/BPS. Disruption of the lining layer of the bladder (known as the urothelium) may cause it to become leaky, allowing toxic substances in the urine to irritate the bladder wall.

Claudins are a family of tetra-membrane spanning proteins that form the structural and functioning core of the tight junctions. An extensive analysis of gene expression reported a ninety-fold upregulation of claudin-2 mRNA levels in biopsies of patients with IC/BPS compared with controls [7]. It was also shown that the expression of claudin-2 in the umbrella cell layer increases the permeability of the urothelium to small ions, triggers an inflammatory process in the bladder mucosa and lamina propria, and increases voiding frequency [8].

Many experts believe that IC/BPS is complex and may be a multi-organ disorder. An appropriate animal model can be helpful for diagnosing and understanding IC/BPS, and assessing possible therapeutic options for people with this syndrome.

One of the models that was proposed is a URO-MCP-1 transgenic mouse model for the IC/BPS. This model was developed through microinjection of fertilized eggs with a 4.9 Kb KpnI-DraIII DNA fragment consisting of the uroplakin II (UPII) gene promoter, an intron sequence, the mouse MCP-1 coding sequence with a secretory element, and a poly A

additional site [9]. The bladder of URO-MCP-1 mice constitutively secretes monocyte che-moattractant protein-1 (MCP-1), a key chemokine that plays an important role in diverse inflammatory and chronic pain conditions including IC/BPS [10]. URO-MCP-1 mice show bladder hypersensitivity and develop bladder inflammation upon intravesical administration of a single sub-noxious dose of lipopolysaccharide (LPS). Along with bladder inflammation, URO-MCP-1 mice exhibit pelvic pain and voiding dysfunction, providing a novel model for IC/BPS research.

Intravesical administration of Gd-DTPA in conjunction with the use of dynamic CE-MRI (DCE-MRI) was developed by our group and validated in a rat pre-clinical model, as well as in a small cohort of IC patients [11–13]. We have developed an *in vivo* MRI test to assess increased bladder urothelial permeability in a pre-clinical rat model following protamine sul-fate (PS) exposure using a dynamic contrast-enhanced magnetic resonance imaging (DCE-MRI) approach [12, 13]. This method involves intravesical administration of a contrast agent, Gd-DTPA, to monitor leakage or uptake of this agent through the bladder wall. The enhanced contrast MR imaging approach was found to detect bladder urothelium leakage of the contrast agent in rat bladder urothelia. The CE-MRI approach can also be used to assess the effect of therapeutic intervention regarding decreased bladder urothelium permeability [13]. The advantage of the DCE-MRI approach is that 2D and 3D regions of the bladder urothelium can be assessed to establish if there are areas of the bladder wall that are more sus-ceptible to bladder permeability alterations. Gd-DTPA MRI contrast agent was also taken up and significantly retained in human IC bladder urothelium, compared to normal control blad-ders, demonstrating the method's potential as a diagnostic tool to help evaluate bladder hyper-permeability alterations in IC patients [11]. The agent was well tolerated by all IC patients tested, indicating this technology could be translated rapidly to the clinic.

The purpose of this study was to assess the proposed preclinical mouse model for IC/BPS for bladder hyper-permeability using a developed DCE-MRI approach, and to visualize the expression of the molecular marker claudin-2 using *in vivo* molecular targeted MRI.

## Materials and methods

### Animals

All animal studies were approved by the Oklahoma Medical Research Foundation (OMRF) Institutional Animal Care and Use Committee (IACUC). Adult female, homozygous URO-MCP-1 mice (20-25g) were used for the study. For all procedures, mice were anesthe-tized with isoflurane (1.5–2.0% with 800–1,000 mL $O_2$). On day 0, lipopolysaccharide (LPS) (1 µg of LPS (E. coli 055: B5, Sigma-Aldrich, St. Louis, MO) in 100 µL phosphate-buffered saline (PBS)) was infused into the urinary bladder via an intravesical catheter. A lubricated (lidocaine jelly, 2%) sterile catheter (24 gauge x ¾ in.) was used to transurethally catheterize each animal. LPS was instilled in the bladder for 10min. LPS was then removed from the blad-der using abdominal pressure and followed by 3 saline flushes (100 µL per flush). This dilute LPS concentration is insufficient to induce permeability in wild-type mice [9]. Saline-treated URO-MCP-1 mice were used as controls. Bladder hyper-permeability MRI assessments were made at 24hrs following exposure to LPS (n = 5 for each group). Sham controls were adminis-tered saline (100µL) instead of LPS.

### Molecular-targeted MR imaging probe

The claudin-2 probe was synthesized by coupling an antibody against claudin-2 (mouse monoclonal antibody (12H12) from ThermoFisher Scientific) to albumin in a Gd-DTPA-albu-min-biotin construct (biotin-albumin-Gd (gadolinium)-DTPA (diethylene triamine

pentaacetate); anti-claudin-2 probe). The macromolecular contrast material, biotin–albumin–Gd-DTPA, was prepared using a modification of the method of Dafni et al. [14]. The biotin moiety in the contrast material was added to allow histological localization. Biotin–BSA–Gd-DTPA was synthesized as follows. A solution of biotin–BSA–Gd-DTPA was added directly to the solution of antibody (anti-claudin-2, 20 μg/mL) for conjugation through a sulfo-NHS (N-succinimidyl-S-acetylthioacetate)–EDC (N-succinimidyl 3-(2-pyridyldithio)-propionate) link between albumin and antibody according to the protocol of Hermanson [15]. Sulfo-NHS was added to the solution of biotin–BSA–Gd-DTPA and EDC. This activated solution was added directly to the antibody (anti-claudin-2, 20 μg/mL) for conjugation. The mixture was left to react for at least 2 h at 25°C in the dark. The product was lyophilized and subsequently stored at 4°C and reconstituted to the desired concentration for injections in phosphate buffer saline (PBS). We have previously used this approach for molecular-targeted MRI probes for *in vivo* VEGF-R2 [16, 17], iNOS [18], and free radical [19–24] detection. The probe (0.1 mg/kg bw; 100 μL diluted in saline) was administered via an intravesical catheter, instilled for an hour, and then flushed with saline, to assess retained uptake of the probe. A non-specific IgG isotype contrast agent (mouse IgG-albumin-Gd-DTPA-biotin) was used as a negative control molecular-targeting agent. The molecular weights of either the claudin-2 probe or non-specific IgG isotype contrast agent are estimated to be 232 kDa. The molecular weight of an antibody is ~150KDa. We estimate that there are 1.3 biotin and 23 Gd-DTPA groups bound to each albumin molecule (~70 kDa) [14].

## Magnetic Resonance Imaging (MRI)

MRI experiments were conducted on a 7 Tesla 30 cm-bore Bruker Biospec MRI system.

The bladders were assessed *in vivo* for hyper-permeability using an assay developed by our group as previously described [12, 13]. Gd-DTPA (a MRI contrast agent) was used to assess bladder wall permeability by intravesical injection into the bladder. A RARE T1 sequence was used for the DEC-MRI study Signal intensity was compared before and 7 min after the Gd-DTPA injection. Permeability was assessed by calculating a percent increase in MRI signal intensity just outside the bladder wall from MR images (5 regions-of-interest per animal).

Fort the mt-MRI, a RARE variable TR sequence was used to detect changes in MRI signal intensities and T1 relaxation maps (as previously done [16–24] resulting from the administration of the mt-MRI probe. Pre- and post-probe administration images were taken to establish specific uptake of the claudin-2 probe, or IgG contrast agent, in the bladder urothelia.

## MRI analysis

MRI signal intensities and T1 values were measured from regions-of-interest (ROIs) within images (ROIs were taken in bladder walls from images and T1 maps, along with corresponding regions in saline animal datasets, as displayed on Paravision (v 5.0, Bruker Biospin)).

## Histology, immunohistochemistry (IHC) and immunofluorescence (IF)

*Ex vivo* bladder urothelium tissue samples, utilizing MRI coordinates from the MR images, were assessed by histology and IHC to detect claudin-2 and the anti-claudin-2 mt-MRI probe targeting the biotin moiety with streptavidin horse radish peroxidase (SA-HRP) [21]. The bladders of each animal were removed, preserved in 10% neutral buffered formalin, and processed routinely. Hematoxylin-eosin staining: tissues were fixed in 10% neutral buffered formalin, dehydrated, and embedded in paraffin. Sections were deparaffinized, rehydrated, and stained according to standard protocols. Several reagents were produced by Vector Labs Inc. (VLI; Burlingame, CA). Histological sections (5μm) embedded in paraffin and mounted on

HistoBond® Plus slides (Statlab Medical Products, Lewisville, TX) were rehydrated and washed in Phosphate Buffered Saline (PBS). The sections were processed using the M.O.M® (Mouse on Mouse) ImmPRESS® Polymer kit, Peroxidase (cat# MP-2400, Vector Laboratories, Burlingame, CA). Antigen retrieval (Antigen Unmasking Solution, Citrate-based, (cat# H-3300, Vector Laboratories, Burlingame, CA) was accomplished via 20-minutes in a steamer followed by 30-minutes cooling at room temperature. Sections were treated with a peroxidase blocking reagent (Bloxall®, cat# SP-6000, Vector Laboratories, Burlingame, CA), followed by M.O.M. IgG blocking reagent to inhibit nonspecific binding. Slides were rinsed in PBS and then incubated for 5 minutes with M.O.M. 2.5% Normal Horse Serum. Claudin 2 Monoclonal Antibody (12H12) (1:50; cat# 32–5600, Invitrogen, ThermoFisher Scientific, Waltham, MA) was applied to each section, from bladders not undergoing mt-MRI (n = 5/group x 2 groups; saline vs. LPS), and following a 30 minute incubation in a humidified chamber at room temperature, sections were washed in PBS, the M.O.M. ImmPRESS reagent was applied according to the manufacturer's directions. Sections for streptavidin horse radish peroxidase (SA-HRP), for the bladders that underwent mt-MRI (n = 5/group x 3 groups; saline vs. LPS administered the claudin-2 probe, or LPS-exposed bladders administered the IgG contrast agent), where the streptavidin targets the biotin moiety of the anti-claudin-2 probe, or the IgG contrast agent, were processed as above, except they were incubated for 60 minutes at room temperature with ready to use Streptavidin, Peroxidase, R.T.U. (cat# SA-5704, Vector Laboratories, Burlingame, CA). Appropriate washes were in PBS. Slides were incubated with Vector® NovaRed® Substrate kit, Peroxidase (HRP) (cat# SK-4800, Vector Laboratories, Burlingame, CA) chromogen for visualization. Counterstaining was carried out with Hematoxylin QS Nuclear Counterstain (cat# H-3404, Vector Laboratories, Burlingame, CA). Appropriate positive and negative tissue controls were used.

To quantitate *ex vivo* claudin-2 expression levels from IHC and SA-HRP levels from mouse bladders administered either the claudin-2 probe or the IgG contrast agent by targeting the biotin moieties, five regions-of-interest (ROIs), captured digitally (20× magnification), were identified in each case. Only areas containing urothelial tissue were analyzed. The number of positive pixels was divided by the total number of pixels (negative and positive) in the analyzed areas. ROIs were analyzed and imaged using Aperio ImageScope (Leica Biosystems, Buffalo Grove, IL).

For IF, paraffin-embedded tissues were sectioned at a thickness of 6μm. Following deparaffinization and antigen retrieval, sections were washed in PBS for 3 min and blocked 30 min in 20% Aquablock buffer (Abcam, MA) in TBS. Mouse anti-Claudin 2 (1:300, Cat: 32–5600, RRID: AB_86980, Invitrogen) antibody and rabbit anti-ZO-1 (1:50, Cat: 61–7300, RRID: AB_2533938, Invitrogen) antibody were incubated at 4˚C overnight. The secondary antibodies goat anti-mouse Alexa Fluor 488 and goat anti-rabbit Alexa Fluor 594 were incubated for 1 hour at RT and followed by DAPI (4',6-diamidino-2-phenylindole dihydrochloride) staining. Negative controls were conducted following the same protocol, but without the primary antibody, and were imaged under the same microscopy settings as experimental slides. Images were taken by using Zeiss LSM 880 confocal microscope (Zeiss, Germany) with 63X objective. The intensity of Claudin 2 expression (corrected total cell fluorescence, CTCF) [25–27] was quantified in 3 images for each animal by using Fiji/ImageJ software. Co-localization coefficients M1 (fraction of claudin-2 overlapping with ZO-1) and M2 (fraction of ZO-1 overlapping with claudin-2) were obtained by using JACoP plugin in Fiji/ImageJ software.

## Trans-Epithelial Electrical Resistance (TEER)

Post mortem the mouse bladders were surgically isolated (between 10 and 11:00 AM) and placed into ice cold Krebs buffer composed of 120 mM NaCl, 6 mM KCl, 1.2 mM $MgCl_2$, 1.2

mM $H_2PO_4$, 2.5 mM, $CaCl_2$, 14.4 mM $NaHCO_3$, and 11.5 mM glucose, aerated with 95% $O_2$−5% $CO_2$. The tissues were opened longitudinally and mounted into the biopsy perfusion chambers (Warner instruments, Hamden, CT). Tissues were bathed in oxygenated Kreb's solution at 37˚C. for 30 min. before experimentation. Permeability was assessed electrophysiologically via measurements of TEER, as discussed previously [13]. To calculate TEER, the potential difference (PD) and short circuit current (Isc) were recorded and TEER was calculated using Ohm's law: $I = \frac{PD}{R}$, where R represents resistance.

### Statistical analysis

Levels of increased MRI signal intensities, TEER and IF were assessed by an unpaired two-tailed Student's t-test. The presence of the molecular-targeted MRI imaging probes (either the claudin-2 probe or the non-specific IgG contrast agent) were compared between LPS- and/or saline-treated URO-MCP-1 mice, and analyzed using one- or two-way ANOVA with multiple comparisons (Tukey's or Sidak's respectively) (GraphPad Prism 7). P-values <0.05 were considered significant.

### Results

Hyper-permeability was assessed by calculating a percent increase in MRI signal intensity just outside the bladder wall from MR images (5 regions-of-interest per animal/day). Fig 1B shows the percent change in MRI signal outside the bladder wall after injection of MRI contrast agent into the bladder (intravesical injection). There was a statistically significant three-fold increase in MRI signal intensity for the LPS-treated URO-MCP-1 mice compared to those treated with saline (p<0.05). In addition, TEER was determined for saline- and LPS-treated URO-MCP-1 mouse bladders *ex vivo*, as further supportive data for bladder urothelial hyper-permeability (Fig 1D). There was a significant ~60% reduction in TEER in the LPS-treated mouse bladder urothelial compared to the saline controls (p<0.01). These results confirm that the bladder becomes hyper-permeable with a single dose of LPS in the URO-MCP-1 model.

Immunohistochemistry (IHC) indicates that levels of claudin-2 are elevated in the LPS-treated mouse bladders (Fig 2), and that increased levels of claudin-2 seem to be associated with urothelium damage (Fig 2Bi).

Immunofluorescence (IF) staining for claudin-2 confirms that LPS-exposed URO-MCP-1 mouse bladders have higher levels of claudin-2 (Fig 3). Corrected total mean fluorescence indicated that there was a significant increase in claudin-2 levels, compared to saline controls (p<0.05) (Fig 3C). We also established that claudin-2 expression is highly co-localized with zonula occludens-1 (ZO-1), also known as tight junction protein-1 (co-localization coefficients M1 = 0.740 ± 0.080, M2 = 0.490 ± 0.102 (n = 8 repeat IF co-localization datasets); where M1 is the fraction of claudin-2 overlapping with ZO-1, and M2 is the fraction of ZO-1 overlapping with claudin-2). Fig 3D, in particular, shows high co-localization in the LPS-exposed URO-MCP-1 mouse bladder.

Fig 4 indicates the feasibility of conducting *in vivo* molecular-targeted MR imaging for claudin-2. Claudin-2 probe levels are significantly increased (p<0.05) in LPS-treated mouse bladders (Fig 4A–4C), compared to saline controls, which had low levels (Fig 4B and 4C). A negative control for molecular-targeting using a non-specific IgG isotype contrast agent (mouse IgG-albumin-Gd-DTPA-biotin), administered to LPS-exposed mouse bladders, indicated no preferential uptake (Fig 4Cii and 4D). The IgG contrast agent administered LPS-exposed mouse bladders were found to be not significantly different in % relative expression of the respective contrast agents, compared to saline-treated mouse bladders administered the claudin-2 probe.

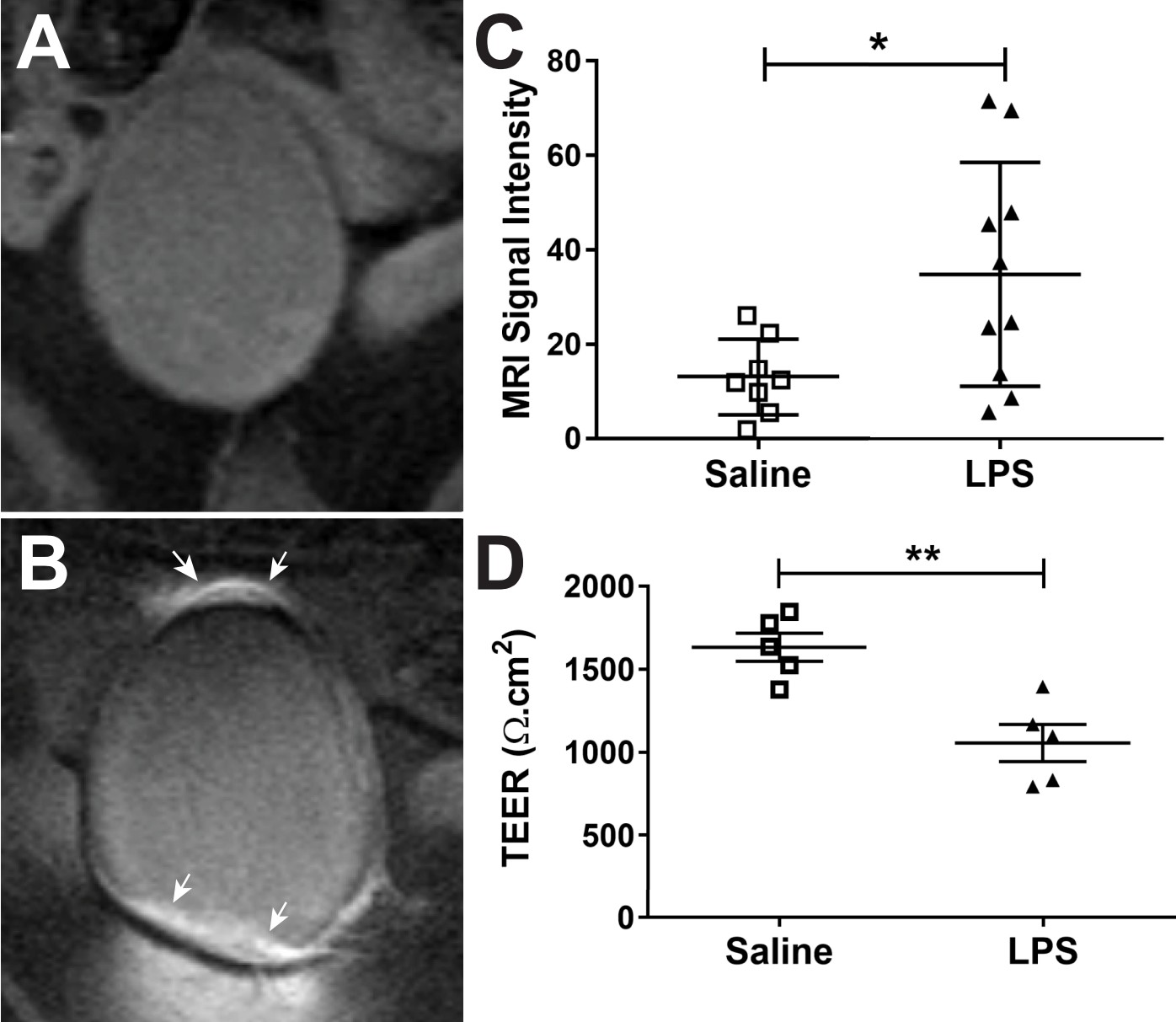

**Fig 1. Bladder hyper-permeability is present in the URO-MCP-1 IC mouse model.** (A, B) Representative CE-MR images of either saline- (A) or LPS-treated (B) URO-MCP-1 mouse bladders. Note hyperintense regions of the bladder in the LPS-treated mouse bladder (white arrows). (C) Percent change in MRI signal intensities outside the bladder wall after intravesical injection of MRI contrast agent, Gd-DTPA, into the bladders of saline- (open squares; n = 8) or LPS-treated (closed triangles; n = 10) mice. There was a significant increase in the percent change in MRI signal intensity in LPS-treated URO-MCP-1 mice, compared to saline-treated mice (*p<0.05). (D) Trans-epithelial electrical resistance (TEER) ($\Omega$.cm$^2$) of mouse bladders (*ex vivo*) that were either treated with saline (open squares; n = 5) or LPS (closed triangles; n = 5). There was a significant decrease in TEER in LPS-treated URO-MCP-1 mice, compared to saline-treated mice (**p<0.01). Statistical analysis involved using an unpaired, two-tailed Student's t-test.

Streptavidin-horse radish peroxidase (HRP) was used to target the biotin moiety of the claudin-2 probe and a non-specific IgG isotype contrast agent in *ex vivo* tissue (Fig 5), which supports the *in vivo* data. The LPS-exposed mouse bladders had significantly higher positivity for SA-HRP, compared to saline controls, whom were both administered the claudin-2 probe (Fig 5D). The IgG negative control mouse bladders exposed to LPS were found to have

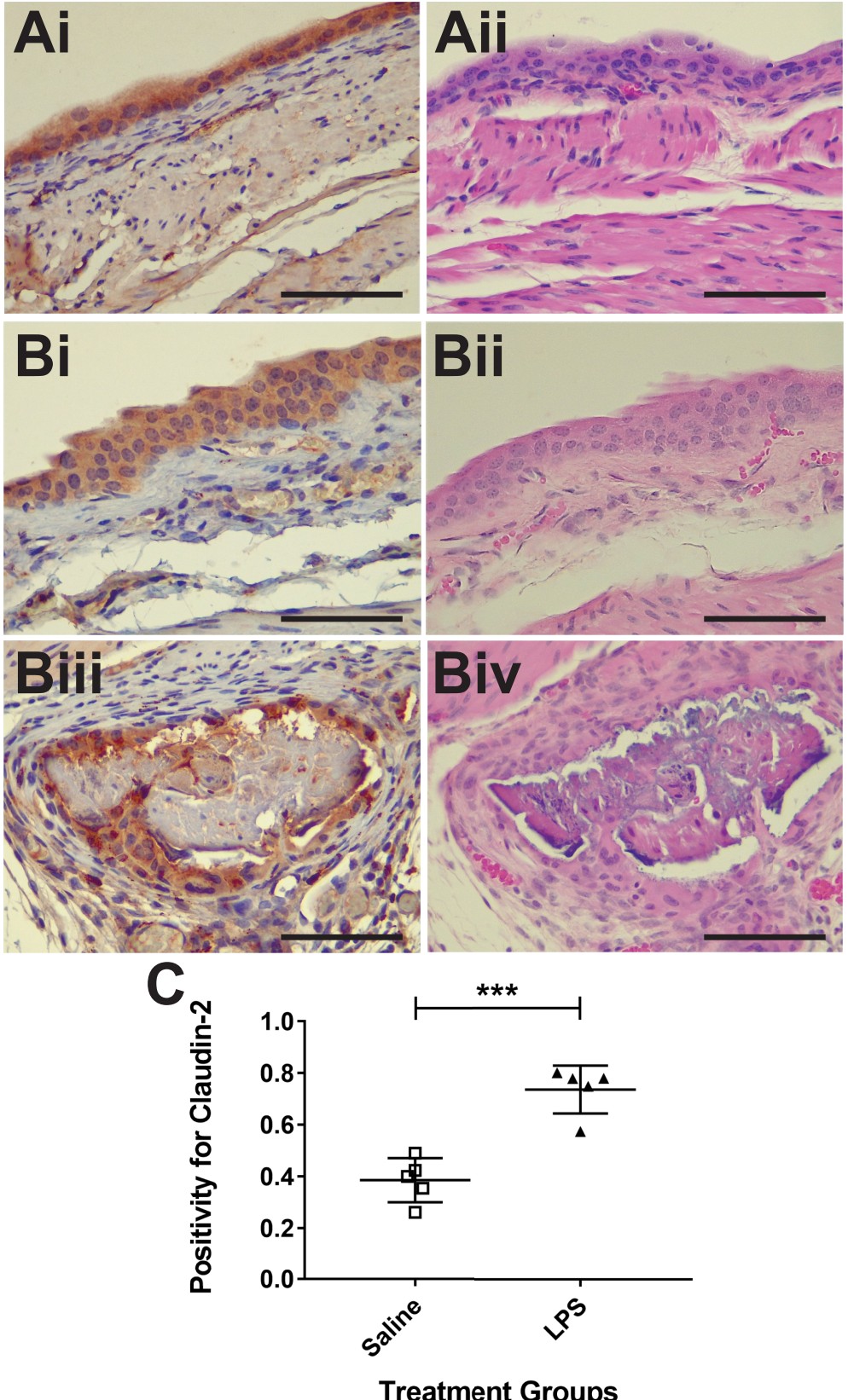

**Fig 2. *Ex vivo* claudin-2 levels detected by immunohistochemistry (IHC) are elevated in LPS-exposed URO-MCP-1 transgenic mouse model for IC.** Representative IHC images obtained from excised *ex vivo* URO-MCP-1 mouse bladders treated with either saline (A) or LPS (B). IHC claudin-2 stained images (i and iii) beside H&E histology images (ii and iv). Note higher levels of IHC staining for claudin-2 in LPS-exposed URO-MCP-1 urothelia within mouse bladders (Bi). All images are taken at 40x magnification (Bars in each frame are 50 μm). (C) Quantification of claudin-2 staining in either saline- (n = 5) or LPS-treated (n = 5) URO-MCP-1 mouse bladder urothelia. Statistical analysis involved using an unpaired, two-tailed Student's t-test.

significantly less positivity for SA-HRP, compared to both the claudin-2 probe administered mouse bladders either treated with saline or LPS (****$p < 0.0001$ for both) (Fig 5D).

## Discussion

For this study, a well-developed transgenic mouse model (URO-MCP-1) was used. In this model, the monocyte chemoattractant protein-1 (MCP-1; also named CCL2) is secreted by the bladder epithelium and develops bladder inflammation, pelvic pain and voiding dysfunction upon intravesical administration of a single sub-noxious dose of LPS. It was previously shown that URO-MCP-1 mice manifest significant functional changes at 24hrs after cystitis induction [9]. These functional changes included pelvic pain as measured by von Frey filament stimulation, and voiding dysfunction (increased urinary frequency, reduced average volume voided per micturition, and reduced maximum volume voided per micturition), as measured in micturition cages [9]. It was previously shown that URO-MCP-1 model demonstrates the symptoms of IC/BPS and could be used as a novel model for IC/BPS research [9]. This study demonstrates the additional manifestation of the disease with an associated increased bladder permeability (see Fig 1). *Ex vivo* TEER was also used to support the *in vivo* MRI hyper-permeability data.

A Gd-DTPA-albumin-biotin construct (as previously used by our group [16–24] was used to conjugate the antibody against the specific permeability biomarker claudin-2 to the albumin moiety. The advantage of this construct involves not only assessing *in vivo* molecular expression assessment of claudin-2, but also allows direct *ex vivo* tissue confirmation regarding whether the molecular imaging probe reached its' intended target.

Claudin-2 has been well documented to increase in cystitis [28]. Claudin-2, is over-expressed in relationship to bladder wall hyper-permeability, particularly in the umbrella cells, and this urothelial barrier dysfunction may trigger bladder inflammation and alter bladder function [8, 28–30]. We found in this study that claudin-2 was increased in the URO-MCP-1 LPS-induced model for interstitial cystitis. In addition, IF images of the mouse bladder urothelial, obtained in this study, supports the notion that claudin-2 plays a role in the tight-junction network [31, 32].

## Conclusions

In this study we established that the URO-MCP-1 transgenic mouse model for interstitial cystitis (IC), in addition to lower pelvic pain and bladder voiding characteristics associated with IC, also has bladder urothelial hyper-permeability, which has been shown to be relevant also in several IC patients. In addition, we also demonstrated that *in vivo* levels of claudin-2, a known bladder urothelial permeability biomarker, can be visualized *in situ* using a molecular-targeted MR imaging approach, and was validated with histological confirmation. The URO-MCP-1 transgenic mouse model for IC can be considered for pre-clinical evaluation of possible therapeutic approaches. Claudin-2 should be considered as a biomarker regarding therapeutic approaches for bladder urothelial hyper-permeability.

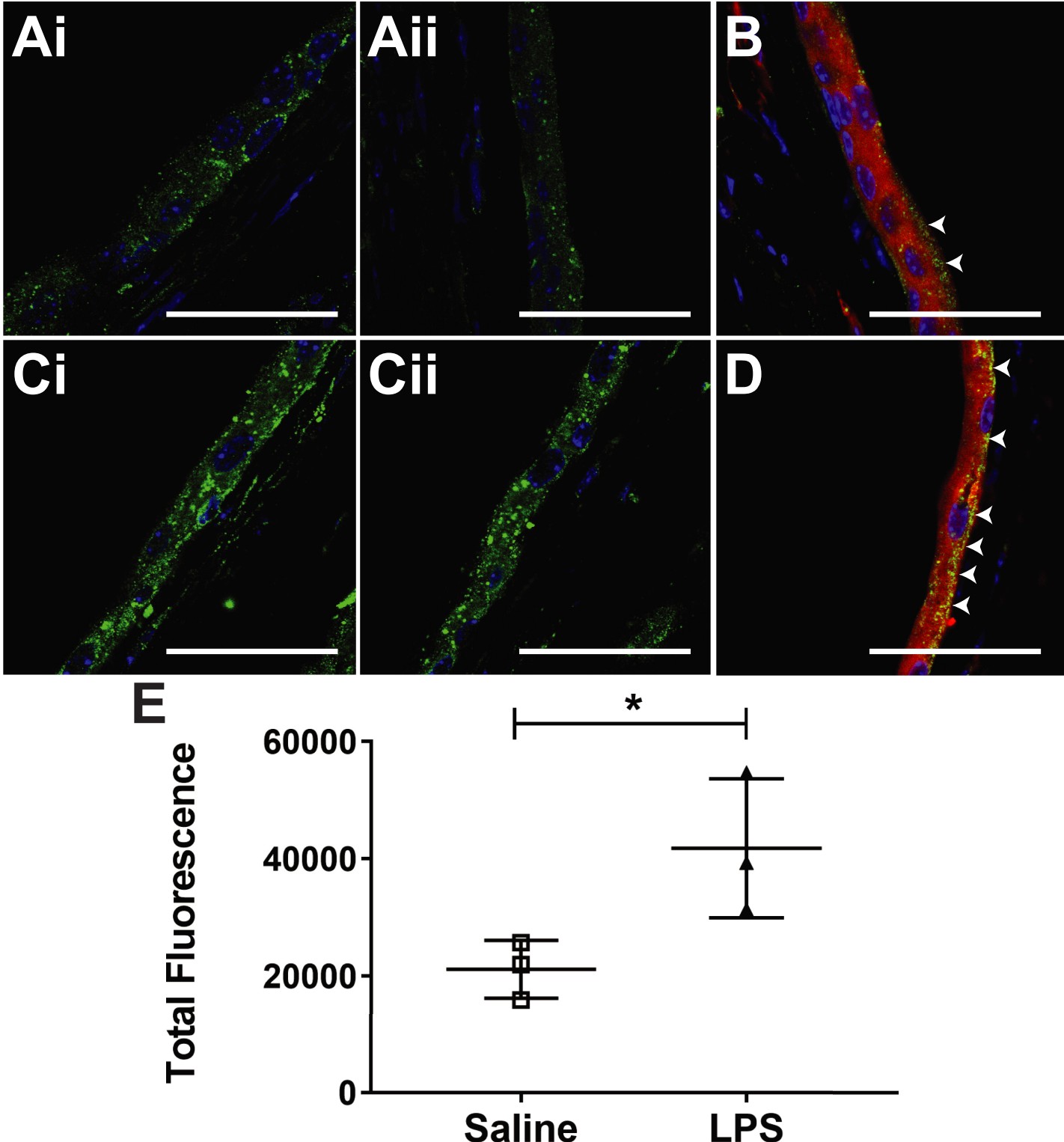

**Fig 3. *Ex vivo* claudin-2 levels detected by immunofluorescence (IF) are elevated in LPS-exposed URO-MCP-1 transgenic mouse model for IC.** Representative IF images obtained from excised *ex vivo* URO-MCP-1 mouse bladders treated with either saline (A) or LPS (C). Representative co-localization IF images of claudin-2 (green) and ZO-1 (red) for either saline (B) or LPS (D)-treated mouse bladders. White arrow heads indicate tight-junction regions with high co-localization areas for both claudin-2 and ZO-1. Claudin-2 was stained with Alexa Fluor 488 (green), ZO-1 was stained with Alexa Fluor 594 (red), and cell nuclei are stained with DAPI (blue). IF claudin-2 stained images (i and ii). Note higher levels of IF staining for claudin-2 in LPS-exposed URO-MCP-1 urothelia within mouse bladders (Ci and Cii). Claudin-2 seems to be detected in the plasma membrane, cytoplasm, per-nuclear, as well as the tight junctions, however co-localization with ZO-1 seems to be

represented mainly in tight junctions (B and D). All images are taken at 63x magnification (Bars in each frame are 50 μm). (E) Quantification of claudin-2 staining in either saline- (n = 3) or LPS-treated (n = 3) URO-MCP-1 mouse bladder urothelia. Statistical analysis involved using an unpaired, two-tailed Student's t-test.

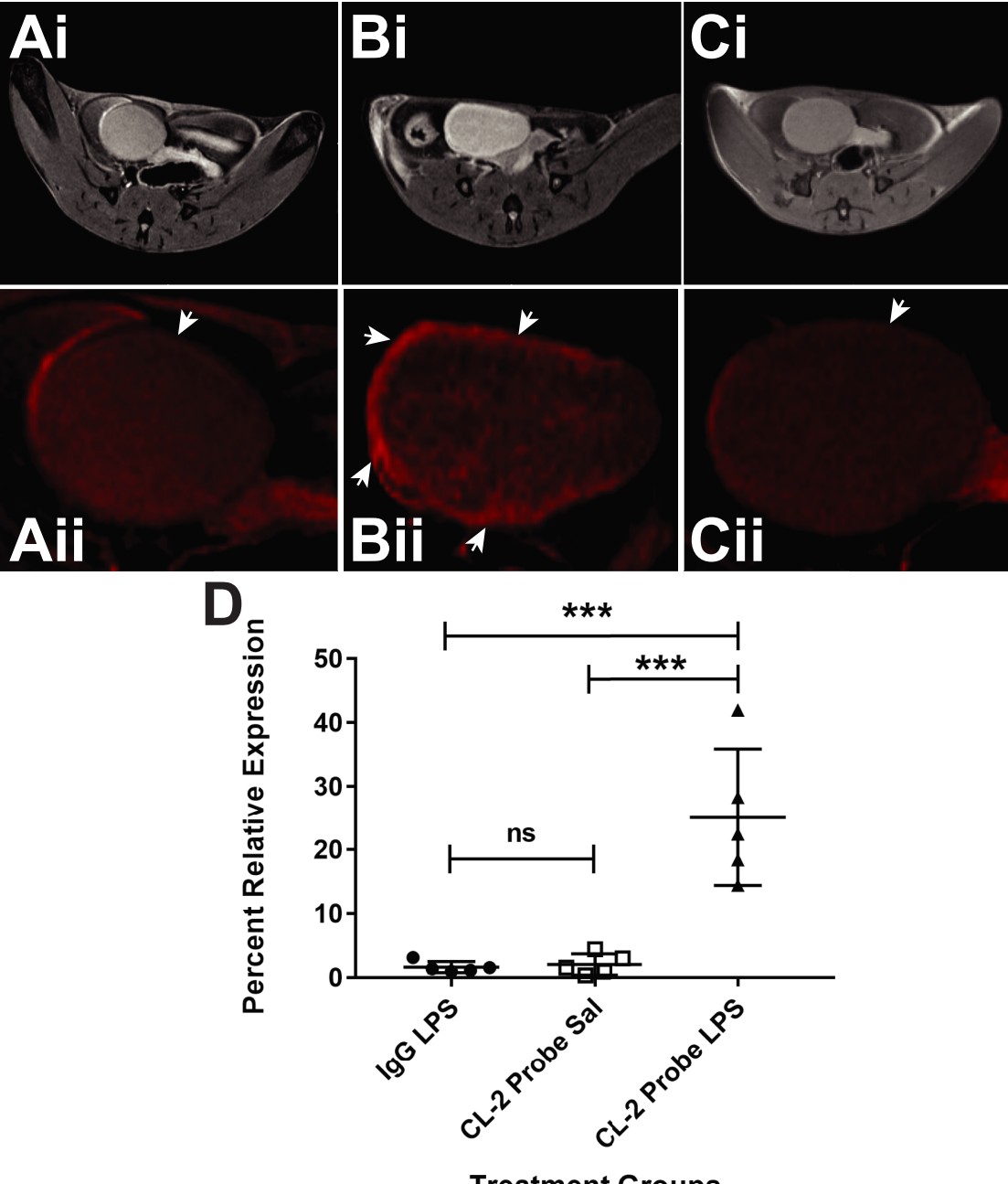

**Fig 4. *In vivo* claudin-2 expression is increased in the LPS-exposed URO-MCP-1 IC mouse model.** (A, B) Representative MR images (i) morphological or (ii) mt-MRI false-colored (red) images of URO-MCP-1 mouse bladders either treated with saline (A) or LPS (B) and the claudin-2 probe, or LPS and the non-specific IgG contrast agent (C). (D) Percent relative expression of Claudin-2, as measured by the percent increase in MRI T1 relaxation changes in either saline- or LPS-treated URTO-MCP-1 mouse bladders administered the Claudin-2 mt-MRI probe. A non-specific IgG contrast agent was used as a negative control in LPS-exposed mice. There was a significant increase in the detection of the Claudin-2 probe in LPS-treated URO-MCP-1 mice, compared to the saline control or LPS mice administered the IgG contrast agent (***p<0.001 for both). Statistical analysis involved using a two-way ANOVA with multiple comparisons (Sidak's).

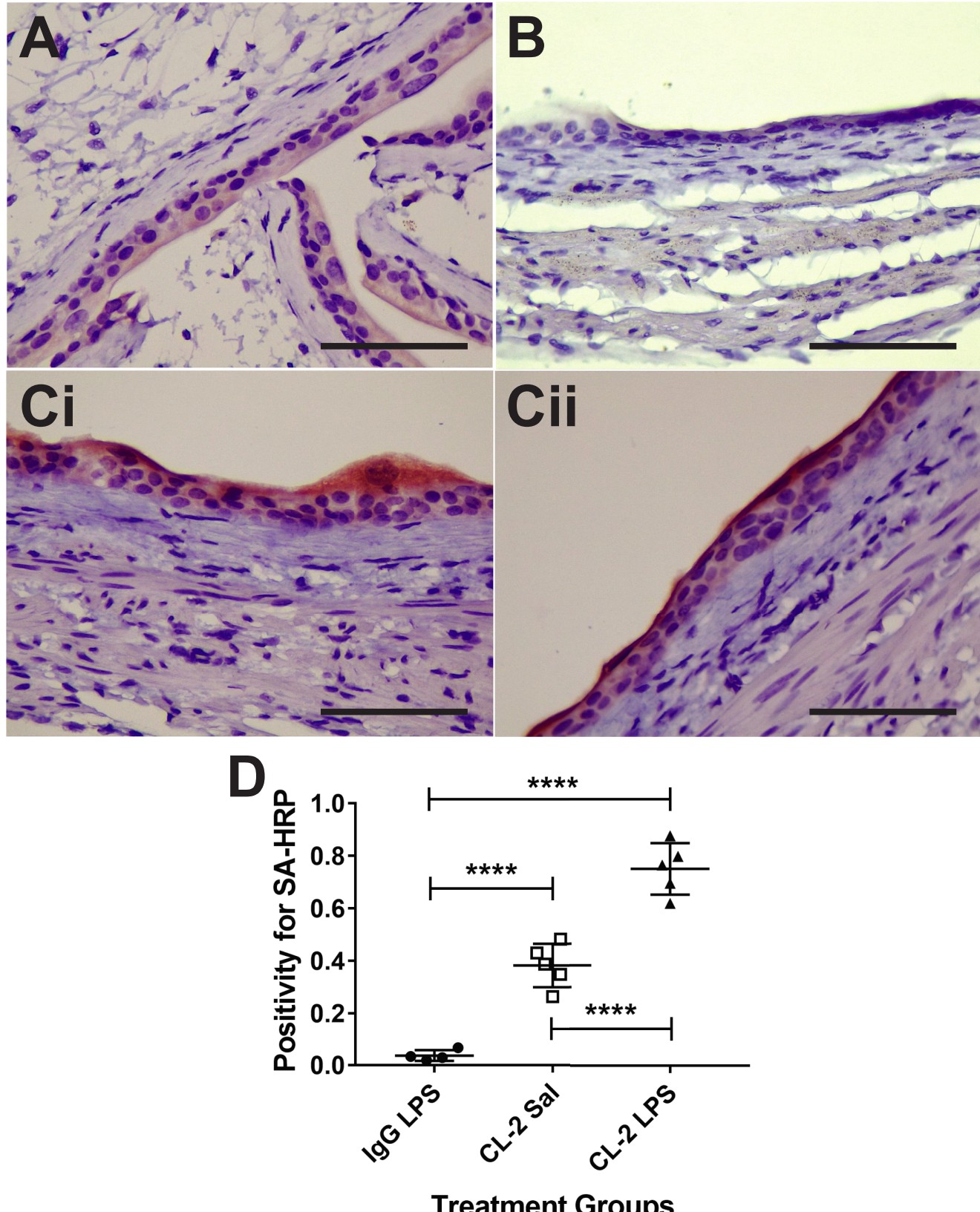

**Fig 5. Verification of claudin-2 molecular-targeted MRI (mt-MRI) probe in LPS-exposed URO-MCP-1 mouse bladders.** Representative images (i and ii) of streptavidin-horse radish peroxidase (SA-HRP) stained mouse bladders targeting the biotin moiety of the anti-claudin-2 mt-MRI probe in saline (A) or LPS (C)-treated mouse bladders. Note higher levels of claudin-2 probe in LPS-exposed mouse bladders (Ci and Cii). (B) Representative image of a SA-HRP stained mouse bladder targeting the biotin moiety of a non-specific mouse IgG contrast agent in an LPS-exposed mouse bladder. All images are taken at 40x magnification (Bars in each frame are 50 μm). (D) Quantification of SA-HRP for either the non-specific IgG contrast agent in LPS-treated (n = 5), or the claudin-2 probe in saline- (n = 5) or LPS-treated (n = 5) URO-MCP-1 mouse bladder urothelia. Statistical analysis involved using a two-way ANOVA with multiple comparisons (Sidak's).

## Supporting information

**S1 File.**
(ZIP)

## Author Contributions

**Conceptualization:** Yi Luo, Robert E. Hurst, Beverley Greenwood-Van Meerveld, Rheal A. Towner.

**Data curation:** Nataliya Smith, Debra Saunders, Megan Lerner, Michelle Zalles, Nadezda Mamedova, Ehsan Mohammadi, Tian Yuan.

**Formal analysis:** Nataliya Smith, Michelle Zalles, Nadezda Mamedova, Daniel Cheong, Ehsan Mohammadi, Tian Yuan, Rheal A. Towner.

**Funding acquisition:** Beverley Greenwood-Van Meerveld, Rheal A. Towner.

**Investigation:** Nataliya Smith, Rheal A. Towner.

**Methodology:** Nataliya Smith, Debra Saunders, Megan Lerner, Michelle Zalles, Ehsan Mohammadi, Tian Yuan, Rheal A. Towner.

**Resources:** Yi Luo.

**Supervision:** Beverley Greenwood-Van Meerveld.

**Validation:** Robert E. Hurst, Rheal A. Towner.

**Visualization:** Megan Lerner.

**Writing – original draft:** Rheal A. Towner.

**Writing – review & editing:** Nataliya Smith, Debra Saunders, Megan Lerner, Michelle Zalles, Nadezda Mamedova, Daniel Cheong, Ehsan Mohammadi, Tian Yuan, Yi Luo, Robert E. Hurst, Beverley Greenwood-Van Meerveld.

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
