## [Decision Letter · Decision Letter 0]

12 Apr 2020

PONE-D-20-05854

In vivo Assessment of Bladder Hyper-Permeability and Using Molecular Targeted Magnetic Resonance Imaging to Detect Claudin-2 in a Mouse Model for Interstitial Cystitis

PLOS ONE

Dear Dr. Towner,

Thank you for submitting your manuscript to PLOS ONE. After careful consideration, we feel that it has merit but does not fully meet PLOS ONE’s publication criteria as it currently stands. Therefore, we invite you to submit a revised version of the manuscript that addresses the points raised during the review process.

This manuscript was carefully reviewed by one reviewer and as a whole the manuscript very well received.  However, the reviewer pointed out some important control experiments which will require significant additional experimentation.  Given the care that the reviewer took in the review, we have decided that a second reviewer will not be needed at this time.  For the manuscript to be acceptable, please address the reviewer critiques as well as the following 6 points.

IHC staining may be acceptable if it can be demonstrated in images that claudin-2 is indeed at the tight junctions.  If not, then IF should be considered because it may give a better signal, as suggested by reviewer.  

Quantification of all IHC and/or IF is required.

For the figure 2 images, please consider whether there is an orientation difference or whether there is a difference in the nature of the umbrella cell layer.  Consider choosing a better representative image if there are not overt histological differences.

Scale bars should be present in all microscopic images.

Check the labeling of Y axes.  If it is percent, label as percent

Avoid the word leaky when discussing claudin 2 and use the word hyperpermeability instead since it is a selectively permeable pathway.

We would appreciate receiving your revised manuscript within 6 months. To enhance the reproducibility of your results, we recommend that if applicable you deposit your laboratory protocols in protocols.io, where a protocol can be assigned its own identifier (DOI) such that it can be cited independently in the future. For instructions see: http://journals.plos.org/plosone/s/submission-guidelines#loc-laboratory-protocols

We look forward to receiving your revised manuscript.

Kind regards,

Christopher R. Weber, MD, PhD

Academic Editor

PLOS ONE

Journal Requirements:

2. We noticed you have some minor occurrence of overlapping text with previous publications, which needs to be addressed:

http://www.painful-bladder.org/pdf/2017-01_Newsletter.pdf

https://link.springer.com/article/10.1007%2Fs00404-017-4364-2

https://www.medicinenet.com/interstitial_cystitis/article.htm

https://doi.org/10.1523/ENEURO.0381-16.2017

In your revision ensure you cite all your sources (including your own works), and quote or rephrase any duplicated text outside the methods section. Further consideration is dependent on these concerns being addressed.

Reviewers' comments:

Reviewer's Responses to Questions

**Comments to the Author**

1. Is the manuscript technically sound, and do the data support the conclusions?

Reviewer #1: Partly

2. Has the statistical analysis been performed appropriately and rigorously? 

Reviewer #1: Yes

3. Have the authors made all data underlying the findings in their manuscript fully available?

Reviewer #1: Yes

4. Is the manuscript presented in an intelligible fashion and written in standard English?

Reviewer #1: Yes

5. Review Comments to the Author

Reviewer #1: The manuscript of Smith and colleagues describes a novel technique to evaluate urothelial permeability and detect claudin-2 using magnetic resonance imaging. This approach was tested in the URO-MCP-1 mouse model of interstitial cystitis (IC), which exhibits increased urothelial permeability. IC diagnosis is based upon symptoms and exclusion of mimicking diseases. Therefore, this method, if validated, could have a significant impact in the diagnosis of this condition. Below are some comments that might help to improve the manuscript.

Major

A major concern is the specificity of the antibody-based probe towards claudin-2. A control with a probe with inactivated antibody, a probe with a control antibody or the biotin-albumin-Gd-DTPA should be performed. Fig. 1 shows that using the Gd-DTPA probe the authors observe change in permeability between controls and LPS treated mice. Because the backbone of the antibody-based probe is Gd-DTPA, the question is whether this probe is binding to claudin-2 or just simply detecting changes in permeability.

Please provide a detailed description of the animal model in the introduction, this will help the reader.

Additional information should be provided about the method for coupling the claudin-2 antibody to the probe and the purification of the probe. Authors are encouraged to provide enough information so that the experiment can be duplicated. How was the mass of the probe determined?

Page 9, line 156. List the reagents produced by Vector Labs. State the concentration of claudin-2 Atb used.

Number of animals and statistical test used should be indicated in the figure legends.

Page 11, line 198. The images showed in Fig. 2 should be improved. It looks that the umbrella cell layer is gone in the LPS treated bladder. Is this correct? If possible, authors should quantify urothelial damage.

Claudin-2 only appears to be expressed in umbrella cells in control and across the whole urothelium in mice treated with LPS. Unfortunately, the immunohistochemical staining does not allow to define the cellular localization of claudin-2, which is relevant for its detection with the antibody-based probe. Authors should consider using immunofluorescence to assess whether Cldn2 is expressed in the plasma membrane and /or tight junctions.

The number in the panels of Fig.2 do not correspond with the figure legend.

Page 13, last paragraph. The facts that the authors observe a signal by immunohistochemistry does not mean that the probe is detecting claudin-2 in the tissue. If the urothelium is leaky, the probe will diffuse and can be covalently attached to the tissue during fixation. This issue should be addressed.

Page 14, line 248. The statement is incorrect. There is no evidence that the overexpression of claudin-2 inhibits the synthesis of tight-junction proteins.

The increase in claudin-2 expression in mice treated with LPS should be quantified by western-blot.

Minor

Page 7, line 114. remove and retained for 10 min.

Please rephrase last sentence of the conclusions.

6. PLOS authors have the option to publish the peer review history of their article (what does this mean?). If published, this will include your full peer review and any attached files.

Reviewer #1: No

---

## [Author Response · Author response to Decision Letter 0]

30 Jul 2020

I would like to thank the Reviewers for their thoughtful suggestions and comments. We have addressed each comment below. The changes that we have made to the manuscript were substantial, and have dramatically improved the quality of the revised manuscript. Major changes include adding immunofluorescence data, adding an IgG isotype negative control for the molecular-targeting imaging data, quantification of all data, expanding the Methods section, and addition of relevant references.

Editor’s comments:

For the manuscript to be acceptable, please address the reviewer critiques as well as the following 6 points.

1. IHC staining may be acceptable if it can be demonstrated in images that claudin-2 is indeed at the tight junctions. If not, then IF should be considered because it may give a better signal, as suggested by reviewer. 

IF was included – new Figure 3.

2. Quantification of all IHC and/or IF is required.

Quantification for every figure has been included.

3. For the figure 2 images, please consider whether there is an orientation difference or whether there is a difference in the nature of the umbrella cell layer. Consider choosing a better representative image if there are not overt histological differences.

Better representative images have been included.

4. Scale bars should be present in all microscopic images.

Added as suggested.

5. Check the labeling of Y axes. If it is percent, label as percent

Changes as suggested.

6. Avoid the word leaky when discussing claudin 2 and use the word hyperpermeability instead since it is a selectively permeable pathway.

Changed.

Reviewer #1: 

1. A major concern is the specificity of the antibody-based probe towards claudin-2. A control with a probe with inactivated antibody, a probe with a control antibody or the biotin-albumin-Gd-DTPA should be performed. 

A non-specific IgG isotype contrast agent (replacing the claudin-2 antibody with a mouse IgG) has been added.

2. Fig. 1 shows that using the Gd-DTPA probe the authors observe change in permeability between controls and LPS treated mice. Because the backbone of the antibody-based probe is Gd-DTPA, the question is whether this probe is binding to claudin-2 or just simply detecting changes in permeability.

The probe is intravesically administered, but then washed out with saline after distilling in the bladder for an hour. Any residual probe would be washed out and not detected. The SA-HRP data clearly shows that the probe is present in the bladder urothelium.

3. Please provide a detailed description of the animal model in the introduction, this will help the reader.

Now included.

4. Additional information should be provided about the method for coupling the claudin-2 antibody to the probe and the purification of the probe. Authors are encouraged to provide enough information so that the experiment can be duplicated. How was the mass of the probe determined?

Additional information on the synthesis of the claudin-2 and IgG contrast agents has been included, along with an estimation of the molecular weight.

5. Page 9, line 156. List the reagents produced by Vector Labs. State the concentration of claudin-2 Atb used.

Updated and included.

6. Number of animals and statistical test used should be indicated in the figure legends.

Now included.

7. Page 11, line 198. The images showed in Fig. 2 should be improved. It looks that the umbrella cell layer is gone in the LPS treated bladder. Is this correct? If possible, authors should quantify urothelial damage.

We have used better representative images for Figure 2, as well as included quantification of Clnd2 expression. We have also additionally added trans-epithelial electrical resistance (TEER) data in Figure 1, that supports the increased permeability/damage to the LPS-induced mouse bladders, compared to saline controls.

8. Claudin-2 only appears to be expressed in umbrella cells in control and across the whole urothelium in mice treated with LPS. Unfortunately, the immunohistochemical staining does not allow to define the cellular localization of claudin-2, which is relevant for its detection with the antibody-based probe. Authors should consider using immunofluorescence to assess whether Cldn2 is expressed in the plasma membrane and /or tight junctions.

Immunofluorescence data has been added in a new figure, including quantification. From IF images it seems that Cldn2 is expressed in both the plasma membrane and tight junctions (added in figure legend – new Figure 3).

9. The number in the panels of Fig.2 do not correspond with the figure legend.

Changed.

10. Page 13, last paragraph. The facts that the authors observe a signal by immunohistochemistry does not mean that the probe is detecting claudin-2 in the tissue. If the urothelium is leaky, the probe will diffuse and can be covalently attached to the tissue during fixation. This issue should be addressed.

We are stating that we detect an increase in streptavidin-horse radish peroxidase (SA-HRP) which specifically targets the biotin moiety of either the claudin-2 probe or the IgG isotype control contrast agent. This can only occur if either probe or contrast agent is present. This has been shown to be the case for several probes using the Ab-albumin-Gd-DTPA-biotin construct by the authors (of which some of these references have been cited in this manuscript). We have included a non-specific IgG isotype contrast agent as a negative control. We have also rephrased the paragraph. 

11. Page 14, line 248. The statement is incorrect. There is no evidence that the overexpression of claudin-2 inhibits the synthesis of tight-junction proteins.

Corrected.

12. The increase in claudin-2 expression in mice treated with LPS should be quantified by western-blot.

Quantification of the IHC, immunofluorescence, and molecular-targeted probe via streptavidin horse radish peroxidase which binds to the biotin moiety of the claudin-2 probe (and the non-specific IgG isotype contrast agent) was evaluated instead. 

Minor

13. Page 7, line 114. remove and retained for 10 min.

Corrected.

14. Please rephrase last sentence of the conclusions.

Changed.

---

## [Decision Letter · Decision Letter 1]

19 Aug 2020

PONE-D-20-05854R1

In vivo and ex vivo Assessment of Bladder Hyper-Permeability and Using Molecular Targeted Magnetic Resonance Imaging to Detect Claudin-2 in a Mouse Model for Interstitial Cystitis

PLOS ONE

Dear Dr. Towner,

Thank you for submitting your manuscript to PLOS ONE. After careful consideration, we feel that it has merit but does not fully meet PLOS ONE’s publication criteria as it currently stands. Therefore, we invite you to submit a revised version of the manuscript that addresses the points raised during the review process.

Please address the concerns of reviewer 1 and also please show additional IF images demonstrating claudin colocalization with ZO-1 or other tight junction protein.  Much of the staining appears cytoplasmic or perinuclear and it is important to show that it colocalizes with the tight junction as well.

We look forward to receiving your revised manuscript.

Kind regards,

Christopher R. Weber, MD, PhD

Academic Editor

PLOS ONE

Reviewers' comments:

Reviewer's Responses to Questions

**Comments to the Author**

1. If the authors have adequately addressed your comments raised in a previous round of review and you feel that this manuscript is now acceptable for publication, you may indicate that here to bypass the “Comments to the Author” section, enter your conflict of interest statement in the “Confidential to Editor” section, and submit your "Accept" recommendation.

Reviewer #1: All comments have been addressed

2. Is the manuscript technically sound, and do the data support the conclusions?

Reviewer #1: Yes

3. Has the statistical analysis been performed appropriately and rigorously? 

Reviewer #1: No

4. Have the authors made all data underlying the findings in their manuscript fully available?

Reviewer #1: Yes

5. Is the manuscript presented in an intelligible fashion and written in standard English?

Reviewer #1: Yes

6. Review Comments to the Author

Reviewer #1: The authors have addressed satisfactorily my previous comments.

An additional comment, t-test should be used for use to compare data from two groups (Fig. 1, 2 and 3). ANOVA is generally used to compare data from more than two groups.

7. PLOS authors have the option to publish the peer review history of their article (what does this mean?). If published, this will include your full peer review and any attached files.

Reviewer #1: No

---

## [Author Response · Author response to Decision Letter 1]

19 Aug 2020

I would like to thank the Reviewers for their thoughtful suggestions and comments. We have changed the wording and included a Student’s t-test for data presented in Figs. 1, 2 and 3. 

Reviewer #1: 

The authors have addressed satisfactorily my previous comments.

An additional comment, t-test should be used for use to compare data from two groups (Fig. 1, 2 and 3). ANOVA is generally used to compare data from more than two groups.

We have updated the statistical analyses and wording for Figs. 1-3 in the Methods and relevant figure legends. The p-values were not substantially changed, i.e. the asterisks reported for each figure remain the same, and therefore no changes were made to the figures.

---

## [Editor Report · Decision Letter 2]

27 Aug 2020

PONE-D-20-05854R2

In vivo and ex vivo Assessment of Bladder Hyper-Permeability and Using Molecular Targeted Magnetic Resonance Imaging to Detect Claudin-2 in a Mouse Model for Interstitial Cystitis

PLOS ONE

Dear Dr. Towner,

Thank you for submitting your manuscript to PLOS ONE. After careful consideration, we feel that it has merit but does not fully meet PLOS ONE’s publication criteria as it currently stands. Therefore, we invite you to submit a revised version of the manuscript that addresses the points raised during the review process.

Please provide IF and/or IHC images with demonstrate claudin-2 protein expression at the tight junction.  It states in the legend that it is at the tight junction, however, presently, it is difficult to discern this.  There is also a lot of cytoplasmic and perinuclear staining, which should also be described in addition to the membranous staining.  It may also help to include an arrow or two pointing to the tight junctions.   

We look forward to receiving your revised manuscript.

Kind regards,

Christopher R. Weber, MD, PhD

Academic Editor

PLOS ONE

---

## [Author Response · Author response to Decision Letter 2]

31 Aug 2020

I would like to thank the Editor and Reviewers for their thoughtful suggestions and comments. We have included co-localization data for claudin-2 and ZO-1, and updated the text in the Abstract, Methods, Results and Figure 3 legend, as a result. 

Editor: 

Please provide IF and/or IHC images with demonstrate claudin-2 protein expression at the tight junction. It states in the legend that it is at the tight junction, however, presently, it is difficult to discern this. There is also a lot of cytoplasmic and perinuclear staining, which should also be described in addition to the membranous staining. It may also help to include an arrow or two pointing to the tight junctions. 

We have included co-localization data for claudin-2 and ZO-1 in Figure 3 and the Results section, which includes quantitative co-localization coefficient M1 and M2 data, as well as the Methods section. We have also updated the wording regarding the IF staining to say, “Claudin-2 seems to be detected in the plasma membrane, cytoplasm, per-nuclear, as well as the tight junctions, however co-localization with ZO-1 seems to be represented mainly in tight junctions (B and D).”

---

## [Editor Report · Decision Letter 3]

3 Sep 2020

In vivo and ex vivo Assessment of Bladder Hyper-Permeability and Using Molecular Targeted Magnetic Resonance Imaging to Detect Claudin-2 in a Mouse Model for Interstitial Cystitis

PONE-D-20-05854R3

Dear Dr. Towner,

We’re pleased to inform you that your manuscript has been judged scientifically suitable for publication and will be formally accepted for publication once it meets all outstanding technical requirements.

Kind regards,

Christopher R. Weber, MD, PhD

Academic Editor

PLOS ONE
---

## [Editor Report · Acceptance letter]

14 Oct 2020

PONE-D-20-05854R3 

***In vivo* and *ex vivo* Assessment of Bladder Hyper-Permeability and Using Molecular Targeted Magnetic Resonance Imaging to Detect Claudin-2 in a Mouse Model for Interstitial Cystitis**

Dear Dr. Towner:

I'm pleased to inform you that your manuscript has been deemed suitable for publication in PLOS ONE. Congratulations! Your manuscript is now with our production department. 

Kind regards, 

on behalf of

Dr. Christopher R. Weber 

Academic Editor

PLOS ONE